# The Role of Antioxidant Plant Extracts’ Composition and Encapsulation in Dietary Supplements and Gemmo-Derivatives, as Safe Adjuvants in Metabolic and Age-Related Conditions: A Review

**DOI:** 10.3390/ph17121738

**Published:** 2024-12-23

**Authors:** Bogdan-Stefan Negreanu-Pirjol, Ticuta Negreanu-Pirjol, Florica Busuricu, Sanda Jurja, Oana Craciunescu, Ovidiu Oprea, Ludmila Motelica, Elena Iulia Oprita, Florentina Nicoleta Roncea

**Affiliations:** 1Faculty of Pharmacy, Ovidius University of Constanta, 6, Capitan Aviator Al. Serbanescu Str., 900470 Constanta, Romania; bogdan.negreanu@univ-ovidius.ro (B.-S.N.-P.); florentina.roncea@univ-ovidius.ro (F.N.R.); 2Academy of Romanian Scientists, 3, Ilfov Str., 050044 Bucharest, Romania; ovidiu.oprea@upb.ro (O.O.); ludmila.motelica@upb.ro (L.M.); 3Faculty of Medicine, Ovidius University of Constanta, 1, University Alley, 900470 Constanta, Romania; sanda.jurja@univ-ovidius.ro; 4National Institute of Research & Development for Biological Sciences, 296, Splaiul Independentei, 060031 Bucharest, Romania; oana.craciunescu@incdsb.ro (O.C.); iulia.oprita@incdsb.ro (E.I.O.); 5Faculty of Chemical Engineering and Biotechnologies, National University of Science and Technology Politehnica Bucharest, 1-7, Polizu Str., 011061 Bucharest, Romania; 6Research Center for Advanced Materials, Products and Processes, National University of Science and Technology Politehnica Bucharest, 060042 Bucharest, Romania

**Keywords:** antioxidants, dietary supplements, nutrivigilance, gemmotherapy, encapsulation, oxidative stress

## Abstract

Given the current global circumstances, marked by severe environmental pollution—including the contamination of food—along with daily stress and a sedentary lifestyle, many consumers choose to improve their quality of life by using, among others, minimally processed food, food supplements, and gemmo-derivatives. Recent lab and clinical studies have shown the positive impact of specific nutrients with antioxidant capacities in the treatment of several conditions generated by oxidative stress. This paper reviews antioxidant plant extracts utilized as components in various dietary supplements and gemmoderivatives, highlighting their chemical composition and biological properties in preventing diseases caused by oxidative stress. A modern approach to food science brings to the fore the concept of dietary supplements vs. functional food, nutraceuticals, and gemmo-derivatives. The definitions of these terms are not being unanimously regulated in this respect and describe each category of compound and product, also emphasizing the need to implement adequate nutrivigilance. In order to enhance the absorption and bioavailability of dietary supplements and gemmo-derivatives based on antioxidant plant extracts, some encapsulation techniques are outlined.

## 1. Introduction

In the current global conditions, the intense pollution of the environment (e.g., carbon-based smoke), exposure to UV radiation, xenobiotics (e.g., drugs, pesticides, carcinogens), daily stress and a sedentary lifestyle are key contributors to oxidative stress and the development of various metabolic disorders, including type 2 diabetes, fatty liver disease, obesity, and heart disease. They also play a significant role in age-related conditions, such as neurodegenerative diseases and eye disorders, including age-related macular degeneration (AMD), diabetic retinopathy, cataracts, glaucoma, and dry eye disease. These health problems are mainly triggered by oxidative stress [1].

Reactive oxygen species (ROS) are generated in our body as a result of metabolic processes and oxidation reactions due to exogenous factors [2]. The process of oxidation in the human body produces unstable free radicals, which damage cellular membranes and other structures [3]. In certain conditions, the endogenous antioxidant enzyme system (superoxide dismutase—SOD; catalase—CAT; glutathione oxidase—GPX), as well as the exogenous supply of antioxidant vitamins and minerals, are insufficient to counteract the effect of oxidative stress [4].

Many consumers choose to improve their quality of life by using, among others, minimally processed foods, dietary supplements, functional food, or gemmo-derivatives with antioxidant capabilities [5,6]. Following cures, a positive impact has been reported in oxidative stress-driven metabolic and age-related conditions [7,8].

In this context, this review aims to explore relevant research on the composition and encapsulation of dietary supplements and gemmo-derivatives based on natural antioxidants from plant extracts, highlighting the mechanisms of action, and focuses on their adjuvant effect demonstrated in clinical trials for animals and patients with metabolic and age-related conditions.

## 2. Dietary Supplements, Authentication, and Nutrivigilance

### 2.1. Dietary Supplements vs. Functional Food

Dietary supplements contain one or more substances with physiological effects (vitamins, minerals, amino acids, enzymes, essential fatty acids, pre- and probiotics, lycopene, lutein, coenzyme Q10, taurine, carnitine, inositol, glucosamine, chitosan, spirulina, soy isoflavones, medicinal plants, and plant extracts), according to recent European regulations approaching those founded by the USA [9,10,11,12]. Currently, it has not been possible to reach a unitary legislation, neither between continents nor in European countries, with flexibility and adaptability as appropriate. Dietary supplements complement the daily diet, but the consumer should not exceed the recommended dose written on the label/package leaflet [13,14]. The decision to use dietary supplements must be taken upon the recommendation and under the supervision of a doctor, pharmacist, or nutritionist, so as to ensure health and quality of life. The choice of a dietary supplement should be made after verification that it has been noted by competent authorities and that the rules of good manufacturing practice and quality standards of the pharmaceutical industry are complied with by the manufacturer. Inadequate consumption, the properties of some active compounds, and drug–dietary supplement interactions can be multiple causes of adverse reactions of these dietary supplements. An analysis of the European Union notes a balance between the share of supplements containing vitamins and minerals (50%) with those containing other substances with nutritional and physiological effects (43%), but it is not harmonized at the legislation level in all countries [15,16].

Despite the fact that dietary food supplements also present risks for health, nevertheless that they are recommended by a specialist, they accumulate many benefits that improve health and quality of life, such as follows: they ensure a quantifiable supply of nutrients (some of which are essential), prevent the installation certain health problems, and are easy to administer (Figure 1).

Functional food, defined for the first time in 1980 as daily consumed, nutrient-enriched food by the Japanese academic society implementing the FOSHU (Food for Specific Health Use) legislation [17]. Unlike dietary supplements, functional food should not be delivered in a pharmaceutical form but as natural food [18,19]. When functional food provides health benefits in preventing and even curing diseases, it is called a nutraceutical [20,21,22], but regulatory authorities do not clearly delineate these notions.

### 2.2. Dietary Supplements’ Authentication

A globalized phenomenon faced by international organizations, which cannot be stopped, is the adulteration of natural dietary supplements through accidental or deliberate falsification using other plants to those declared, contamination with pesticides, mycotoxins, molds, heavy metals, dioxins, or nitrites, or the use of fillers [23]. Traditional methods of plant identification involve botanical taxonomic analysis, macroscopic and microscopic examination, and chemical methods. They can often give erroneous results, especially when they are applied to intensively processed plants, making morphological characterization impossible [24]. Recent modern methods are those of detecting species-specific DNA sequences, i.e., DNA barcodes, as official methods of the botanical recognition of a plant [25]. Despite all the regulations and authentication methods, the marketing of mislabeled, falsified, and counterfeit dietary supplements persists in commercial markets all over the world [26], but the fines are much smaller compared to the profits.

### 2.3. The Concept of Nutrivigilance

The pharmacovigilance system is defined as “the system for reporting suspected adverse reactions, necessary for the protection of public health, allowing the prevention, detection and evaluation of reactions” by the Directive 2010/84/EU and EU Regulation no. 1235/2010 [27]. A medicinal product must be effective and safe for the consumer with as few as possible side effects. Other adverse reactions than those listed in the package leaflet for consumer information need to be urgently reported to the European Medicines Agency (EMA) and the Scientific Committee for Pharmacovigilance Risk Assessment (PRAC).

The increased consumption of dietary supplements/nutraceuticals leads to unknown risks for consumers, as a result of many possible interactions between the supplements, food, and drugs [28]. Similarly to pharmacovigilance regulations regarding the use of drugs and medical devices, it is necessary for authorities to implement “Nutrivigilance” regulations for the safe use of dietary supplements/nutraceuticals [29]. In recent years, few countries (USA, France, Czech Republic) have introduced nutrivigilance regulations to provide safe dietary supplements/nutraceuticals and to detect changes in the benefit–risk ratio. In this regard, the implementation of nutrivigilance was achieved by the French Agency for Food, Environment, Health and Safety Optional (ANSES), established in 2009 [30].

The World Health Organization (WHO) has adapted the definition of nutrivigilance from the definition of pharmacovigilance, as “science relating to the detection, evaluation, understanding and prevention of adverse reactions produced by common foods, food supplements, novel foods, fortified foods, including energy drinks, medicinal plants and algae” [31]. In addition, it is desirable to configure a unitary nutrivigilance scheme not only at the European level but also at the global level, since agencies are beginning to receive more reports on the adverse effects of dietary supplements [32]. For instance, it has been noted that dietary supplements containing spirulina may be contaminated with cyanotoxins, bacteria, or trace metals. As a result, the ANSES advised against their consumption by individuals with phenylketonuria or those with a predisposition to allergies [30]. It was also reported that dietary supplements containing monacolin K presented the risk of interactions with anticholesterolemic drugs or with some foods (grapefruit), leading to muscle and liver damage [30]. The ANSES’s nutrivigilance classification highlighted the negative health effects of consuming energy drinks, primarily due to caffeine, especially when combined with taurine and glucuronolactone [33]. There are many data in the literature that draw attention to the uncontrolled consumption of energy drinks among young people, especially if combined with alcohol, having cardiovascular or neurological disorders as consequences (anxiety, nervousness, panic attacks, hallucinations, sleep disorders) [34,35]. Many energy drinks contain a series of synthetic additives (preservatives, dyes, acidity correctors) that decrease the quality and safety of the product, especially when they act synergistically, leading to a negative effect. Thus, the potentiation of ADHD syndrome has been demonstrated as a result of the use of synthetic dyes (tartrazine, orange, yellow, blue) in combination with sodium benzoate [36].

Due to a balance between the health benefits and adverse reactions of dietary supplements/nutraceuticals [37], there is the need for their regulation and control through experimental evaluations or clinical trials. In this regard, the Romanian Association of Medicine, Dietary Supplements and Medical Device Manufacturers (RASCI) initiated the MedAccess campaign, promoting that dietary supplements/nutraceuticals should be managed responsibly [38].

## 3. Gemmotherapy Extracts (Gemmo-Derivatives)

### 3.1. Gemmo-Derivatives’ Definition

A special category of dietary supplements are gemmotherapy extracts or gemmo-derivatives. Starting from Dr. Pol Henry’s definition that all informative energy is located in the meristematic tissues of both trees and herbaceous plants, recent approaches emphasize new concepts targeting the biological and quantum connections that state that gemmotherapy’s systemic actions ensure the improvement of intercellular communication, according to the body’s needs, through psycho-neuro-endocrine–immunological mechanisms, correcting the energy behavior of the altered tissue, and thus, the biochemical reactions can be restored to normal [39,40]. As stated, gemmotherapy restores the energy–informational matrix of the body by activating energetic harmony between the compartments of the involved systems (immune, vascular, and nervous systems and mesenchymal stem cells) and, at the same time, by stimulating the recovery processes of damaged and senescent tissues/cells. From this point of view, gemmotherapy is not considered an allopathic therapy but as a future high-performance therapy, an integrative one, taking into account the restoration of cellular communication, the energetic, functional and morphological recovery of the affected tissue, and, finally, that of the organism and a lifestyle in harmony with the environment [41,42] (Figure 1).

Phytoembryotherapy is considered Dr. Pol’s method, based on the “glycerol macerated concentrated” made by the cold maceration of embryonic tissues in water, alcohol, and glycerin without dilution to obtain products named glycerin macerates (GMs). Gemmotherapy is based on the glycerin macerate 1DH, i.e., liquid preparations obtained without water, just with alcohol and glycerin, and diluted 10 times as homeopathic preparations, as described in the European Pharmacopoeia monograph, current edition (Figure 2). GMs’ preparation is also described in the French and Italian Pharmacopoeias, similarly to homeopathic preparations described in the current edition of the European Pharmacopoeia.

In the present day, the use of concentrated GMs (C-GMs), which are undiluted to 1DH, is spreading. C-GMs are about 10 times more concentrated than GMs and, therefore, are used at lower doses (adults: about 5–15 gtt/day) than those used for diluted extracts (adults: up to 1 gtt/kg/day) [43]. At the national and international level, gemmotherapy is practiced with GM and C-GM products that are already on sale and not standardized in active principles responsible for the therapeutic action. Their potential for healing a series of diseases is claimed based on traditional medicine and clinical evidence, although no relationship has been established between the effect, on one side, and their composition and properties, on the other side [43,44,45].

### 3.2. Gemmo-Derivatives’ Preparation

In order to obtain reproducible C-GMs, it is necessary to introduce standard procedures in their cultivation, harvest, and post-harvest treatments [46]. The leaflet associated with a gemmotherapy extract should present its chemical composition and that of the plant from which it was obtained [47].

There are two conventional extraction methods most commonly used for obtaining gemmotherapy extracts (Figure 3). The first one involves the maceration of fresh buds in a solution of water, alcohol, and glycerol, in which water participates in the extraction of hydrophilic active components and reduces the alcohol and glycerol concentration in the final products. The second method involves maceration in an equal volume of alcohol and glycerol, and the only water present is that contained in the meristematic tissues [43].

Taking into account the six principles of green extraction introduced by Chemat et al. (2012) [48] and the twelve principles of green chemistry set by the Environmental Protection Agency of the USA, eco-compatible extraction techniques for obtaining gemmotherapy products aim to reduce their environmental impact, in terms of time and energy, with respect to conventional methods [49]. Ultrasound-assisted extraction (UAE) together with microwave-assisted extraction (MAE), supercritical fluid extraction (SFE), and pulsed electric fields (PEF) are considered modern “green” extraction technologies [50]. Being proved to be efficient, sustainable, low-cost techniques, they reduce both the quantity of employed solvents, preferring alternative solvents (water or food-grade solvents), and the generation of waste, hazardous substances, and consequently pollution.

Most gemmotherapy extracts are obtained from the meristematic tissues of both trees and herbaceous plants (buds or young sprouts). Thus, bud derivatives (BDs), which are commercialized in the European community as plant food supplements according to the Directive 2002/46/EC of the European Parliament, represent a new category of natural herbal products [9]. BDs are glycerin macerates obtained by the traditional method described in the European Pharmacopoeia, maceration, in the presence of a mixture of food-grade solvents, i.e., glycerol and ethanol [51]. There are no health claims for BDs approved by the European Food Safety Authority (EFSA).

### 3.3. Standardization of Gemmo-Derivatives and Biological Role in Metabolic and Age-Related Conditions

BDs’ use as dietary supplements for human health has been reported in traditional medicine for a long time, and *in vitro* and *in vivo* biological studies from the scientific literature support their role as adjuvants in several diseases, based on their chemical composition [52]. Complete analytical fingerprints of gemmo-derivatives represent a useful tool in their standardization. In the context of the FINNOVER cooperative project, the first web application, GEMMAPP (https://www.mdpi.com/2076-3417/13/15/8679/xml, accessed on 27 July 2023) was designed and proposed as a new data repository and tool for providing scientific knowledge about twenty-five botanical species, the recognition of vegetable buds, and descriptions of the main tree species at the base of the most typical commercial BDs of the France–Italy Alcotra territory [53,54].

Due to the large quantity of bioactive compounds identified in gemmotherapy extracts, many of which act synergistically, it is preferable to assess the pharmacological effect of the “phytocomplex” (a combination of different substances, both active principles and other plant components), rather than of any single active compound, as in the case of standard medicine [47]. As for the chemical composition analysis of the phytocomplex and the bioactive components, many conventional techniques are used, like UV-VIS spectrophotometry or modern chromatographic methods, such as thin-layer chromatography (TLC), high-performance liquid chromatography (HPLC) coupled to a diode-array detector (DAD), and gas chromatography (GC) coupled to mass spectrometry (MS) [54].

Berries contain a plethora of antioxidant bioactive compounds, like phenols, anthocyanins, vitamins, and minerals and can be used in the form of GMs to significantly contribute to the treatment of different metabolic and age-related conditions driven by oxidative stress [55,56,57]. The chemical composition of solid-phase microextracted blackcurrant (*Ribes nigrum*) buds indicated the following compounds: terpinolene (15.8%), terpinen-4-ol (33.5%), α-methylbenzenepropanamine (13.2%), spathulenol (15.9%), and caryophyllene oxide (31.1%), according to GC-MS analysis [58]. These volatile compounds defined the tropism of blackcurrant C-GMs for inflammatory diseases, showing that the concentrated extract could significantly counteract the pro-inflammatory response. The reported data supported the traditional use of blackcurrant C-GMs as an immunomodulatory agent of the inflammatory response in oxidative stress-driven diseases.

According to the “multi-marker approach”, total bioactive compound content is determined as the sum of the most important classes of bioactive compounds present in a sample [59]. Four polyphenolic classes are considered: benzoic acids (ellagic and gallic acids), catechins (catechin and epicatechin), cinnamic acids (caffeic, chlorogenic, coumaric, and ferulic acids), and flavonols (hyperoside, isoquercitrin, quercetin, quercitrin, and rutin). Monoterpenes (limonene, phellandrene, sabinene, γ-terpinene, and terpinolene), organic acids (citric, malic, oxalic, quinic, succinic, and tartaric acids) and vitamin C (ascorbic and dehydroascorbic acids) are also considered to obtain a complete analytical fingerprint.

The specific botanical fingerprint of blackcurrant bud preparations from three different phenological stages (bud sleeping, bud break, and first leaves) was assessed by HPLC for the identification and quantification of the phytocomplex’s chemical composition [60]. The phytocomplex from the blackcurrant bud preparations showed the prevalence of organic acids (50.98%), polyphenols (29.39%), monoterpenes (14.04%), and vitamins (5.98%). The three-dimensional “fingerprint” developed for the blackcurrant buds’ derivatives using the multidimensional analytical strategy of excitation–emission fluorescence matrices coupled with HPLC defined their phytochemical profile, and revealed content of quercetin and quercitrin (30–50 mg/100 g fresh weight (FW)), ellagic acid (70 mg/100 g FW), catechins (60–100 mg/100 g FW), and cinnamic acids (5–20 mg/100 g FW) [61]. The high polyphenol content of the blackcurrant buds’ derivatives were endowed with antioxidant and anti-inflammatory activities, useful as adjuvants in ophthalmic and skin diseases [53].

Comparative analyses showed that the fruit extract compositions varied according to the stage of development, season, variety, and extraction method. Thus, the chemical fingerprint of the blackcurrant bud GMs established by HPLC-DAD assay showed three cinnamic acids (chlorogenic acid, caffeic acid, ferulic acid), one flavonol (quercetin), one benzoic acid (gallic acid), and two catechins (catechin, epicatechin). Statistically significant differences were reported between two varieties (Rozenthal and Daniels), with a minimum value of a total phenolic content of 427 mg/100 g FW (Rozenthal 2011 variety) and a maximum of 834 mg/100 g FW (Daniels 2012 variety) [62]. There were also significant differences within the same variety harvested in two different years in regard to the chlorogenic acid concentration, ranging from 4 mg/100 g FW (Rozenthal 2012 variety) to 292 mg/100 g FW (Rozenthal 2011 variety). In Daniels cultivars (years 2011 and 2012), chlorogenic acid was not detected. Also, there were statistically significant differences in catechin and epicatechin between the commercial bud preparations of two different companies, reaching a maximum value of a total phenolic content of 856 mg/100 g FW, similar to the mean value of the Daniels 2012 variety [62]. An *in vitro* microbiological test evidenced the antimicrobial activity of the blackcurrant bud extracts against *Staphylococcus aureus*, *Escherichia coli*, *Pseudomonas aeruginosa*, *Bacillus subtilis*, *Proteus vulgaris*, *Shigella flexneri*, *Klebsiella pneumonia*, and *Enterococcus faecalis* [63].

In the case of the cold maceration of blackcurrant bud derivatives, the chemical fingerprint established by HPLC analysis showed the prevalence of two main groups of bioactive compounds, monoterpenes (82.94%) and catechins (9.46%), over cinnamic acids (3.64%), flavonols (2.67%), and benzoic acids (1.29%) [61].

A quick screening of the best extraction conditions was conducted by UV-Vis and fluorescence coupled with chemometrics for untargeted polyphenolic fingerprints, followed by optimization using the design of the experiment method [59]. The polyphenolic fraction of the optimized pulsed ultrasound-assisted extraction from blackcurrant buds was quantified by a targeted HPLC fingerprint, compared to blackcurrant bud GMs [53]. In the blackcurrant bud GMs, catechins were the most important bioactive class (44.36%), followed by flavonols (27.82%), benzoic acids, and cinnamic acids (19.93% and 7.89%, respectively), while in the PUAE extracts, the quantitative relationship between catechins and flavonols were reversed, with 29.81% for catechins and 42.39% for flavonols, while cinnamic acids (6.20%) and benzoic acids (21.60%) showed percentages similar to the bud macerates [53].

High-resolution fingerprints of blackcurrant bud C-GMs were established using combined HPLC coupled to mass spectrometry analysis and thin-layer chromatography profiles [62]. The phytochemical analysis of the blackcurrant C-GMs led to the identification of 57 compounds distributed in eight chemical classes: flavonoids, nucleosides, phenolic acids, gallotannins and galloyl flavonol glycosides, glycosylated dihydrochalcones, lignans, quinones, and abietane-type diterpenes [61]. Some compounds, such as dihydrochalcone 19, were defined particularly in the blackcurrant [61].

A standardized blackcurrant gemmotherapy extract with a specific phytoconstituent profile contained approximately 133 phytonutrients, such as carboxylic acids, amino acids, vitamins, alkaloids, esters, and terpenes, and significant quantities of flavonoids, such as luteolin, quercetin, apigenin, and kaempferol, according to HPLC–ESI-MS analysis [64]. *In vitro* data obtained in an LPS-induced inflammatory experimental model showed that the extract possessed anti-inflammatory and neuroprotective properties, mainly due to its specific flavonoid content, indicating its potential to be used in complementary therapeutic approaches [64].

The biological functions of blackcurrant buds, which contain high concentrations of vitamin C (107 mg/g) and essential amino acids, as well as their glycerin hydroalcoholic extracts, have been shown to exhibit anti-inflammatory effects and cortisone-like activity. [65]. Di Vito et al. (2020) analyzed the *in vitro* antioxidant, antimicrobial, and immunomodulatory properties of C-GMs obtained from blackcurrant, in comparison to hornbeam (*Carpinus betulus*), fig (*Ficus carica*), and black alder (*Alnus glutinosa*), evaluating their potential in healing upper airway disease [43]. The data of MTT assay supported the cytocompatibility and integrated use of all tested C-GMs. The antioxidant and antimicrobial activities were demonstrated by DPPH and broth microdilution assay, respectively, using ten strains of *Streptococcus pyogenes* and ten probiotic strains. ELISA results exhibited the good immunomodulatory activity of the blackcurrant extracts.

The application of spectroscopic and HPLC fingerprints of extracts from *Vitis vinifera* buds enabled to significantly differentiate between traditional preparations obtained as GMs and innovative extracts obtained by the UAE technique, considering phenolics potential markers of *Vitis vinifera* herbal products. Flavonols were the most abundant class in the ultrasound-based extracts (45%), while phenolic acids were the most important class in the traditional macerates (49%) and commercial bud preparations (about 50%) [66]. HPLC profiles indicated significant quantities of catechin (150 mg/100 g FW) and epicatechin (50 mg/100 g FW) but also other main phenolics (e.g., flavonols, benzoic acids, cinnamic acids) as markers of potential health-promoting activity. Thus, the immunostimulatory extract of the *Vitis vinifera* buds was the only gemmotherapy extract acting as a remedy for acute, chronic, and recurrent inflammatory states, through lymphocytosis in cells involved in the non-specific immune response in the first line of defense [44]. The polyphenols found in the extract have been shown to offer health benefits for metabolic and age-related human conditions [66].

A HPLC chemical fingerprint of raspberry (*Rubus idaeus*) bud GMs identified 26 compounds, which were selected as the main bioactive markers. The chemical profile showed the prevalence of the bioactive class represented by organic acids (63.31%), followed by polyphenolic compounds (25.70%), monoterpenes (10.45%), and vitamins (0.54%) [67]. Due to this composition being rich in flavonoids, raspberry-based gemmotherapy remedies could improve metabolic disorders linked to perimenopausal syndrome [68].

GMs obtained from young shoots of European blueberry (Vaccinium myrtillus), mountain cranberry (Vaccinium vitis-idaea), and blackcurrant were analyzed through LC-MS for phytochemical profiles [69]. *In vitro* studies indicated no cytotoxicity and significant antioxidant activity in human Caco-2 intestinal cells. Biological drainage at the cortical suprarenal and suprarenal levels for the blackcurrant bud GMs and the effect of immunitary and lymphatic biological drainage for the *Vitis vinifera* bud GMs proved to be efficient in improving juvenile rheumatoid arthritis symptomatology [70].

The gemmotherapy extract obtained from the young shoots of blueberry (*Vaccinium myrtilus*) contained polyphenols, flavonoids, and arbutoside evaluated by TLC, HPLC, and UV-VIS spectrophotometry [69]. The presence of polyphenols and flavonoids could reduce oxidative stress, explaining their beneficial effect in cases of retinal complications from diabetes [71]. The high concentration of arbutoside in the extract of cranberry shoots indicated a positive effect in urinary infections, cysts, and colibacillosis [72]. HPLC analysis of a hydroalcoholic extract of blueberry pomace showed a composition rich in phenolics and flavonoids [73]. Moreover, its combination with polysaccharides from chia (*Salvia hispanica*) seeds showed synergistically inhibitive action on α-amylase activity at a higher level (1.36 times) than that of the hydroalcoholic extract, thus demonstrating potential to manage the obesity risk in diabetes.

## 4. Mechanisms of Redox and Inflammation Control

ROS, such as the superoxide radical (O_2_^−^), the hydroxyl radical (OH), singlet oxygen (^1^O_2_), hypochlorite (OCl^−^), non-free radical species of hydrogen peroxide (H_2_O_2_), and ozone (O_3_), and reactive nitrogen species (RNS), such as peroxynitrite (ONOO^−^) and NO, are responsible for oxidative stress, as a main factor driving the pathophysiology of several metabolic and age-related conditions [5,74,75]. The mechanisms of natural antioxidants, such as polyphenols, carotenoids, vitamins, and minerals, administered in the form of dietary supplements and gemmo-derivatives are numerous and still under investigation (Figure 4). Their direct activity targets the scavenging of free radicals through hydrogen atom transfer (HAT) and single electron transfer (SET) mechanisms [76]. Also, natural antioxidants could act indirectly by the stimulation of the endogenous antioxidant system through the interaction and activation of the Nrf2 signaling pathway, in relation to NF-κB inhibition and HO-1 gene expression [77]. Their action is similar in the metabolic conditions driven by oxidative stress, such as diabetes, neurodegeneration, and aging.

One of the main metabolic conditions is type 2 diabetes characterized by hyperglycemia and the development of excessive levels of ROS and advanced glycation end products, in relation to chronic inflammation and tissue damage [78]. As a result of inflammatory cytokine action, the NF-κB family of transcription factors is activated [79]. Significant amelioration has been reported after natural antioxidant intake, as presented in lab and clinical research. Thus, polyphenol-based dietary supplements could inhibit digestive enzymes’ activity, lowering glucose absorption at the intestinal level and also protecting β-pancreatic islet cells by reducing their apoptosis [80]. Berries’ polyphenol extracts inhibited ROS production and reduced lipid accumulation in a model of stressed hepatocytes [81]. Dietary supplementation with vitamin C, α-lipoic acid, and N-acetylcysteine was shown to reduce the systemic side effects of type 2 diabetes through their action on the enzymatic and non-enzymatic antioxidant system [82]. Moreover, natural antioxidants lowered the progression of type 2 diabetes through several mechanisms of preventing lipid peroxidation chain propagation and mitochondrial oxidative stress [83]. Other dietary supplements based on plant extracts rich in polyphenols (quercetin, apigenin, kaempferol, luteolin) have shown potential to scavenge free ABTS radicals and to reduce lipid peroxidation in a liposome model *in vitro* [84].

An *in vitro* study demonstrated that polyphenols also had neuroprotective activity by a NO-induced oxidative stress decrease at the neuronal level and the inhibition of amyloidogenic proteins’ assembly [85]. Small molecules (flavonoids), like quercetin and myricetin, presenting high permeability through the cell membrane due to their size have been recommended as dietary supplements for antioxidant protection in neurodegenerative conditions [86]. Resveratrol is potent antioxidant acting through hydroxyl radical scavenging and metal chelating activity but also by the modulation of cellular pathways involved in the activation of the endogenous antioxidant system [87]. The interaction of flavonoids from dietary supplements with metal ions was significantly modulated by the electron density, pH, and solvent polarity and influenced their membrane permeability and physiological activity, with a higher free radical scavenging capacity of flavonoid–metal complexes being reported, compared to free flavonoids [88]. Endogenous and exogenous ROS action activates a defense response in the human body, defined as inflammation. In an experimental *in vitro* model of LPS-stressed RAW 264.7 macrophages, resveratrol acted as an inhibitor of inflammation biomarkers, COX-1 and COX-2, and NO production [89].

Ophthalmic conditions driven by oxidative stress are associated with ROS action and the damage of DNA, lipids, and proteins, as the main macromolecules of the cell structure [1]. Endogenous ROS induce high retinal metabolism required to convert light into electrical signals sent to the brain, while exogenous ROS are generated by light’s interaction with oxygen on the outer eye surface and passing through the eye to the retina. Being key players in the cell survival of ocular tissues, ROS represent the specific target of alternative medical approaches based on dietary supplements and gemmo-derivatives with antioxidant capabilities [90,91]. Thus, vitamins could diminish ROS-regulated transcription factors, such as NF-κB and HIF1-α, preventing cell autophagy and apoptosis in the cornea and trabecular meshwork [1].

## 5. Clinical Trials

Preclinical animal studies and human clinical research are typically carried out through a multi-step process, which includes identifying bioactive compounds based on their functions and determining the appropriate doses [92]. Regulatory documents set permissible consumption levels and effective doses of bioactive compounds, e.g., taurine, curcumin, or β-glucans in functional food [93]. Standardization following quantum theory was recently approached [94]. Some relevant clinical trials showing the role of antioxidant supplements and functional foods in several metabolic and age-related conditions are summarized in Table 1.

In the case of the metabolic condition diabetes mellitus, clinical trials have shown an important role of antioxidant supplements and functional foods in the prevention of its complications [100]. The systemic side effects generated by hyperglycemia and induced oxidative stress take place through different mechanisms, such as the production of advanced glycation end products, activation of protein kinase-C, and increase in polyol and hexosamine pathways [100]. However, clinical results recommended the intake of multiple antioxidants as a more efficient treatment than that based on a high dose of one antioxidant to avoid its harmful pro-oxidant activity favored by the absence of other antioxidants [100]. Resveratrol has decreased insulin levels in patients with diabetes mellitus due to its antioxidant potential, regulating PPAR-γ and sirtuin 1 expression analyzed in peripheral blood mononuclear cells [95].

Similar mechanisms were described after *in vivo* studies on diabetic retinopathy [101]. Thus, a recent study demonstrated that hyperglycemia promoted the acylation of NF-κB and stimulation of protein-3, inducing the apoptosis of retinal gangliocytes and pericyte loss [102]. Moreover, diabetes activated the expression of MMP-9 regulated by H-Ras and increased MMP-9 activity in the retina of patients with diabetic retinopathy [103], requiring antioxidant strategies based on plant extracts to inhibit ROS production or to stimulate the endogenous antioxidant system. The powerful resveratrol could provide the modulation of miRNAs expression and SIRT1-related pathways, while curcumin had an effect on histone modification, protecting the retina against inflammation [101]. The treatment of selenite-induced cataract disease in mice using antioxidant lutein and baicalin revealed a delay in disease progression [87].

In a recent data analysis of retinal scans of volunteers with dry age-related macular degeneration (AMD), researchers found that taking a daily supplement containing antioxidant vitamins (A, E, beta-carotene) and minerals (zinc, copper) decreased its progression, helping patients with a late-stage condition to preserve their central vision. Trials using supplements substituting the antioxidants lutein and zeaxanthin for beta-carotene improved the efficacy of the treatment and eliminated certain risks [96]. A similar study on healthy subjects (controls) and patients of 50–60 years of age with known degenerative retinal pathology receiving xanthophyll pigment-based supplements showed that the progression of abnormal degenerative vision loss declined to a rate comparable to physiological aging-related vision loss [97]. For subjects over the age of 60 years, the dietary supplements intake was even more effective, compared to the younger group, providing a better control of degenerative processes [97].

Gemmotherapy based on antioxidant bud extracts shows great promise in treating metabolic and ocular conditions through several mechanisms of oxidative stress control. In that way, *Ficus carica* bud extracts reduced the effects of alloxan-induced type 1 diabetes in rats. A reduction in oxidative stress and liver fibrosis in diabetic mice was reported after *Corylus avellana* gemmotherapy extract modulated the secretion of MMPs/TIMP and inhibited the TGF-β1/Smad signaling pathway [98]. GMs of *Vaccinium myrtillus* and *Ribes nigrum* demonstrated antioxidant activity and a capacity to inhibit the lipid peroxidation chain [99]. Also, neuroinflammation induced by lipopolysaccharide injection in adult Wistar rats was attenuated by pretreatment with *Ribes nigrum* gemmotherapy extract, and the pro-inflammatory cytokine TNF-α remained at normal levels in serum, preventing microglial body swelling in the hippocampus [64]. While these mechanisms highlight the valuable potential of gemmo-derivatives, challenges remain in optimizing their efficacy and safety for widespread clinical use.

## 6. Encapsulation Techniques for Safe and Efficient Delivery of Dietary Supplements and Gemmo-Derivatives

### 6.1. Microencapsulation Techniques

To enhance the bioavailability of plant extracts in dietary supplements and gemmo-derivatives, various encapsulation methods have proven effective. Encapsulation can be performed in organic particles, ranging from polymeric particles made of alginate, chitosan, or cellulose to lipid nanocarriers (LNC) or solid lipid nanoparticles (SLN). Methods like emulsion and liposomal entrapment are effective for encapsulating fruit extracts in dietary supplements, enhancing their bioavailability and stability for commercialization. Likewise, spray-drying encapsulation is a promising method for enhancing the bioavailability of fruit extracts in dietary supplements, offering effective encapsulation, characterization, and potential for improved gastrointestinal digestion, health benefits, and controlled release properties [104] (Figure 5).

Microencapsulation (a particle size between 0.5 µm and 1000 µm, depending on the fabrication technique) is a method that preserves bioactives and allows for targeted release, crucial for maintaining nutrient integrity during processing [105]. The encapsulation of plant extracts in dietary supplements enhances bioavailability by the effective delivery of antioxidants, vitamins, and other bioactive compounds, improving stability and nutrient absorption. Modern encapsulation technologies can lower the size of particles to the colloidal domain (<1 µm) but needs nanotechnology to manipulate the bioactive substances at a nanometric size to ensure proper encapsulation and control their release. For example, encapsulating fruit extracts in alginate microcapsules could enhance bioavailability by protecting polyphenols, improving antioxidant activity, and masking unpleasant flavors in dietary supplements. Also, the encapsulation of fruit extracts using prebiotic biopolymers enhances bioavailability in dietary supplements by controlling the release of bioactive compounds, improving their physicochemical and thermal properties. Fractioned cellulose using biopolymer complexation technology is highly efficient for encapsulating fruit extracts, like pomegranate (with a loading capacity up to 20% and a loading efficiency of 85%), in dietary supplements, enhancing their bioavailability through an interpenetrating polymeric network. Utilizing emulsion systems improves the dispersibility and stability of hydrophobic nutraceuticals, enhancing their absorption in the gastrointestinal tract [106].

Polymer-based encapsulation using food-grade biopolymers, like whey protein isolate and chitosan, have been shown to effectively encapsulate nutraceuticals, ensuring their stability and controlled release in the gut [107]. Also, it was reported that the encapsulation of hydrophobic and hydrophilic nutraceuticals in liposomes protected by biopolymers enhanced the stability and controlled bioavailability of bioactives in the gastrointestinal tract. Microspheres made of chitosan and (2-hydroxypropyl) methylcellulose (HPMC) demonstrated the efficient encapsulation of gemfibrozil and effective modulation of the drug release rate. The optimal polymer/drug ratio was found to be 2:1 or 3:1, with low-molecular-weight chitosan yielding the best results [108]. In the case of macromolecular conjugates, the covalent linking of gemfibrozil to polymers like PHEA and PHPA showed varied release profiles, with PHPA providing sustained release over 10 h [108].

Lipid-based nanocarriers are particles with sizes between 10 and 1000 nm, in which active plant components are dissolved, entrapped, or adsorbed (Figure 6). In general, the lipids’ presence enhances the bioavailability of the entrapped components, improves the long-term stability, and ensures a better control of release kinetics. The crystallization degree of the lipid and the dispersion degree and surface of the particles are the factors that influence the bioactive molecules’ release from the carriers. Additional advantages are a high loading capacity, the protection of incorporated compounds, and the ease of manufacturing. The principal disadvantage is the low capacity to load hydrophilic compounds (~2–6%).

The main techniques proposed for lipid nanocarrier preparation are high-pressure homogenization, ultrasonication, solvent evaporation, solvent emulsification–diffusion, and microemulsion methods. From the encapsulation phase, the SLN can be obtained as a solid solution, with an enriched shell or enriched coreLong-term stabilization and storage as colloidal dispersions are achieved when the zeta potential exceeds 30 mV [109].

These encapsulation techniques not only protect plant extracts from degradation but also improve their functional properties, making them more effective in promoting health benefits as antimicrobial agents [110,111]. Thus, the encapsulation of bioactive components in dietary supplements and gemmo-derivatives enhances the stability, bioavailability, and absorption of nutrients in the gastrointestinal tract, improving immunity, well-being, and disease prevention. While encapsulation methods show great promise, challenges remain in regard to optimizing the formulations for different antioxidant plant extracts and achieving optimal release profiles and stability across various conditions. Further research is needed to refine these techniques for broader applications to maximize their health benefits and consumer acceptance.

### 6.2. Nanoencapsulation Techniques

Nanoencapsulation has shown significant potential in enhancing the effectiveness of plant extract-based dietary supplements by improving their stability, bioavailability, and controlled release. This innovative approach utilizes nanotechnology to protect sensitive compounds, thereby maximizing their health benefits.

Encapsulation in mesoporous silica (MCM-41, MCM-48, SBA-15, FDU-12), be it simple or functionalized to tune the release curves, can protect plant extracts from premature oxidation, a low gastric pH, or other factors that could alter bioactive compounds. The extracts can be successfully encapsulated in such mesoporous structures with the direct result of improving biological properties, such as antimicrobial activity [112]. Four mesoporous silica systems based on MCM-41 and MCM-48, raw and functionalized with (3-aminopropyl)triethoxysilane (APTES), with particle sizes of 200–300 nm and pores of ~2 nm, were developed and loaded with plant extract in an innovative method of combining and integrating natural extracts with nanotechnology. The vacuum-assisted method was the preferred approach for loading mesoporous supports, as it ensured that the loading predominantly took place within the pores (Figure 7).

In order to improve the bioavailability of trans-ferulic acid, encapsulation in hexagonal and cubic mesoporous silica support functionalized by the soft-template method using APTES as a modifying agent was conducted [113]. The loaded bioactive molecules exhibited strong interactions with the functionalized support and the release rate was tuned for simulated gastric fluids and simulated intestinal fluids, allowing the sustained drug delivery of biologic active compounds. As a result, the functionalized mesoporous materials could be employed as controlled release systems for antioxidant plant extracts.

The antioxidant plant extracts could also be loaded onto nanoparticles with low porosity by grinding in a one-pot step (Figure 8) [113]. This process could enhance the adsorption of bioactive compounds on the particles’ surface by improving the contact.

Nanoencapsulation enhances the stability of fruit extracts, as demonstrated with *Capparis spinosa*, where encapsulation improved the stability of antioxidant phenolic compounds during storage [114]. The encapsulation process can also increase the bioavailability of active compounds, allowing for better absorption in the body. For instance, papaya leaf extract encapsulated in chitosan showed a notable anti-inflammatory effect, indicating enhanced bioactivity [115]. The nanoencapsulation of grape and apple pomace phenolic extract in chitosan and soy protein via nanoemulsification can enhance the efficacy of antioxidant fruit extract-based dietary supplements by improving their bioavailability and stability [116]. The nanoencapsulation of bioactive flavonoid using the nanoprecipitation technique enhances the effectiveness of fruit extract supplements by improving the bioavailability, shelf life, and generation of nutraceuticals [117]. Even the presence of some phospholipids like phosphatidylethanolamine or phosphatidylserine, the antioxidant activity of α-tocopherol from plant oils can be greatly increased [118].

Nanoemulsions, such as those derived from exotic parijoto fruit extract (*Medinilla speciosa*), facilitate the controlled release of bioactive compounds, which can mitigate adverse effects and improve therapeutic outcomes [119].

The encapsulation process enables precise nutrient targeting, thereby enhancing their effectiveness in functional foods. While nanoencapsulation offers numerous advantages, challenges remain regarding the stability of nanostructures in complex food systems and the potential impact on the biological activity of encapsulated compounds.

### 6.3. Challenges and Limitations of Encapsulation

The development of dietary supplements and gemmo-derivatives faces several significant challenges and limitations, including resistance, pharmacokinetics, and delivery mechanisms, which hinder the effectiveness of the novel therapeutic agents. The health benefits of dietary supplements and gemmo-derivatives are also limited by a number of factors, which are taken into account for their technological design and need to be improved: solubility, degradation, bioavailability, metabolism, and efficacy.

Resistance mechanisms include alterations in dietary supplements’ and gemmo-derivatives’ metabolism and signaling pathways, necessitating the design of new carriers that can overcome these barriers [120]. The encapsulation of supplements and gemmotherapy products employs various methods to enhance the stability and release profiles of therapeutic agents. This process involves techniques that protect bioactive compounds from degradation, improve their solubility, and facilitate their controlled release. The main techniques include drug delivery systems based on polymeric microspheres and macromolecular conjugates, as well as advanced encapsulation technologies in liposomes, solid lipid nanoparticles, nanomicelles, or in situ gelling systems [121]. For instance, site-specific nanotechnology-based systems have been shown to improve the immune response, addressing oxidative stress and inflammation caused by lipid metabolism dysregulation in eye diseases, particularly age-related macular degeneration (AMD) [121].

In regard to pharmacokinetics, bioactives from dietary supplements and gemmo-derivatives have a short plasma half-life and are rapidly metabolized, requiring high doses that can lead to toxicity. Their hydrophilic or hydrophobic nature limits passive diffusion across cell membranes, making them reliant on nucleoside transporters, which can contribute to resistance. Innovative delivery systems, including biodegradable nanocarriers and liposomal formulations, offer potential for facilitating the sustained release of therapeutic agents, improving bioavailability and targeting precision, and reducing systemic side effects [122,123]. According to the mechanisms of bioactives release, dissolution, diffusion, water penetration, and chemically controlled delivery systems have been defined [124]. Zero-order pharmacokinetics allow a constant level of bioactive release from a controlled delivery system, unlike fluctuations, compared to the minimum effective concentration in the case of conventional delivery systems [125]. Clinical testing has presented the long-term and non-invasive technology of the third generation of nanoparticle-based systems, showing their capacity to selectively deliver bioactives to specific sites [126]. However, controlled targeted delivery remains a challenge.

Other challenges in developing encapsulated dietary supplements and gemmo-derivatives include delivery mechanisms. It was shown that the encapsulation of berry anthocyanins improved resistance and controlled release to reach an intended gastrointestinal site [123]. Clinical trials have revealed the benefits of the encapsulation of bioactives and offered solutions in their bioavailability, targeted delivery, and the prevention of adverse reactions [127]. For instance, liposomal polyphenol-rich fractions derived from berries demonstrated health-promoting effects in infected mice, owing to improved solubility and absorption compared to their non-encapsulated counterparts, ultimately resulting in enhanced efficacy [127].

The limitations of bioactives’ encapsulation regard their toxicity and safety applications in nanomedicine. Studies have indicated liver or lung inflammation, neurotoxicity, and immunomodulatory effects after nanocarrier uptake by the reticuloendothelial system or crossing the blood–brain barrier [128,129]. Regulatory guidelines should be approached to delineate the benefit–risk ratio and provide the safe use of encapsulated products for health promotion [130].

While these challenges are still significant, novel research on the encapsulation and delivery systems of dietary supplements and gemmo-derivatives might provide pathways to enhance their efficacy, potentially leading to the improved condition of patients.

## 7. Conclusions

Antioxidant plant extracts and bud gemmo-derivatives have been widely used in traditional medicine. The distinct content of phenolic compounds in buds makes them a category of important botanicals, but they have still been poorly studied. The absence of comprehensive scientific data and clear, unified regulations increases the risk associated with these products, making them susceptible to accidental errors in botanical species identification, as well as to fraud and adulteration. Nutrivigilance regulations are needed.

The polyphenol content of antioxidant plant extracts and bud gemmo-derivatives is strongly influenced by the manufacturing processes whose parameters are often not strictly defined (e.g., solvent ratios in extraction mixtures, raw material/extraction mixture ratios, extraction time) and, thus, they affect their final compositions.

The encapsulation of antioxidant plant extracts and gemmo-derivatives significantly enhance the shelf life of food products by protecting bioactive compounds from degradation and ensuring their controlled release. The technologies address the challenges posed by environmental factors that typically compromise food quality. For example, encapsulation in liposomes improves the solubility and stability of antioxidant compounds, preventing their degradation, and allows the sustained presence of active ingredients, which is crucial for maintaining food safety over extended periods. Other encapsulation systems using biopolymers could facilitate the targeted release of bioactive compounds, ensuring their effectiveness throughout a product’s shelf life, while controlled delivery minimizes the risk of spoilage due to microbial growth. Thereby, the extended usability of food products and promising results in enhancing the antioxidant and functional properties of food are expected, further prolonging shelf life.

While encapsulation offers numerous benefits, it is essential to consider potential challenges, such as the stability of encapsulated systems over time, drug resistance, pharmacokinetics, and delivery mechanisms, which may affect its long-term efficacy in food preservation or the improvement of patients’ conditions.

## Figures and Tables

**Figure 1 pharmaceuticals-17-01738-f001:**
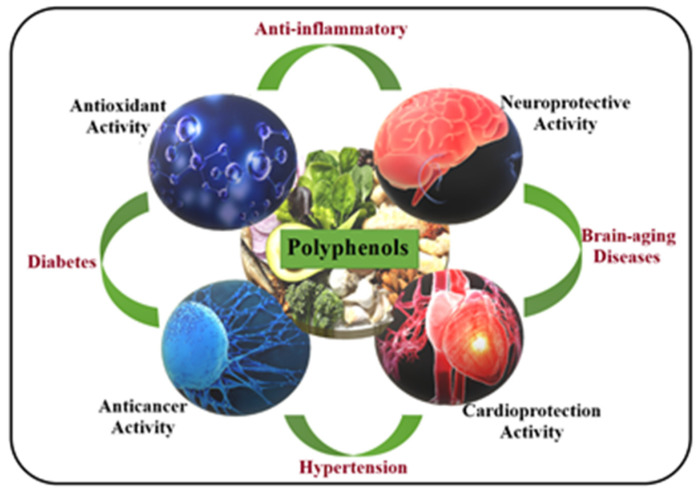
Biological activities of some dietary food supplements.

**Figure 2 pharmaceuticals-17-01738-f002:**
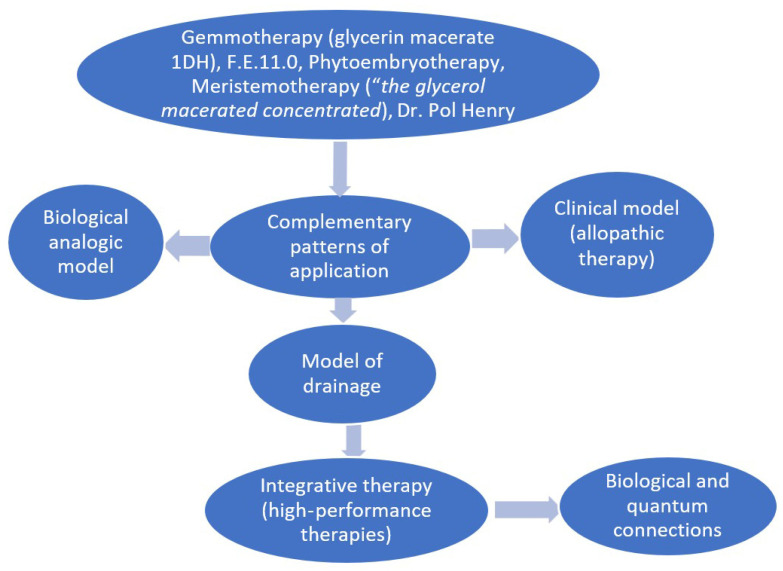
Gemmotherapy depiction and its patterns of application.

**Figure 3 pharmaceuticals-17-01738-f003:**
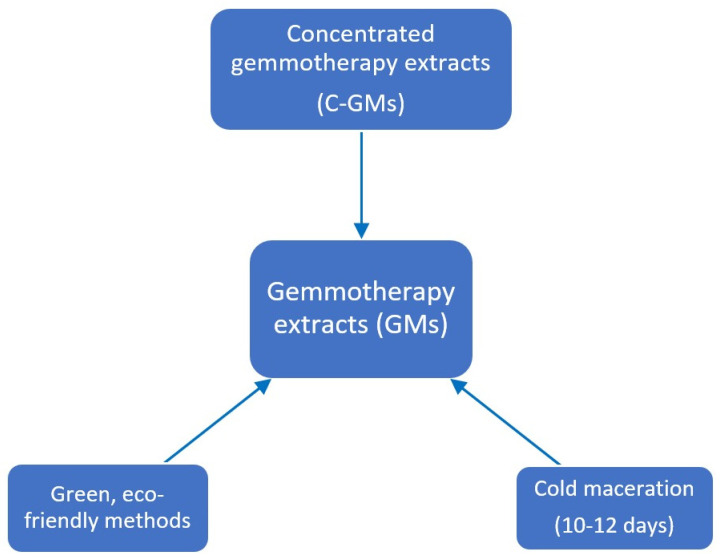
Conventional and green methods used for obtaining gemmotherapy extracts.

**Figure 4 pharmaceuticals-17-01738-f004:**
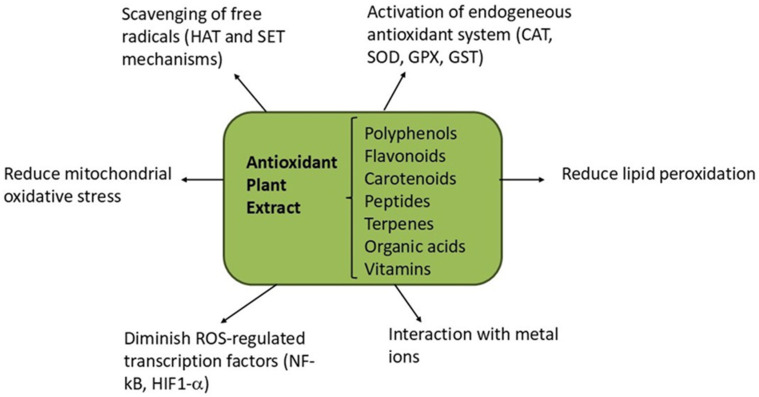
Antioxidant mechanisms of plant extracts administered in form of dietary supplements and gemmo-derivatives.

**Figure 5 pharmaceuticals-17-01738-f005:**
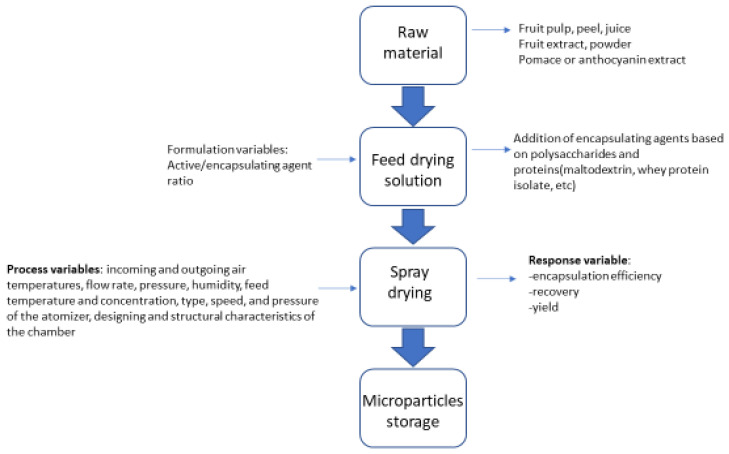
The microencapsulation of antioxidant bioactive compounds from fruits by spray drying, highlighting the variables of the process (adapted from [104]).

**Figure 6 pharmaceuticals-17-01738-f006:**
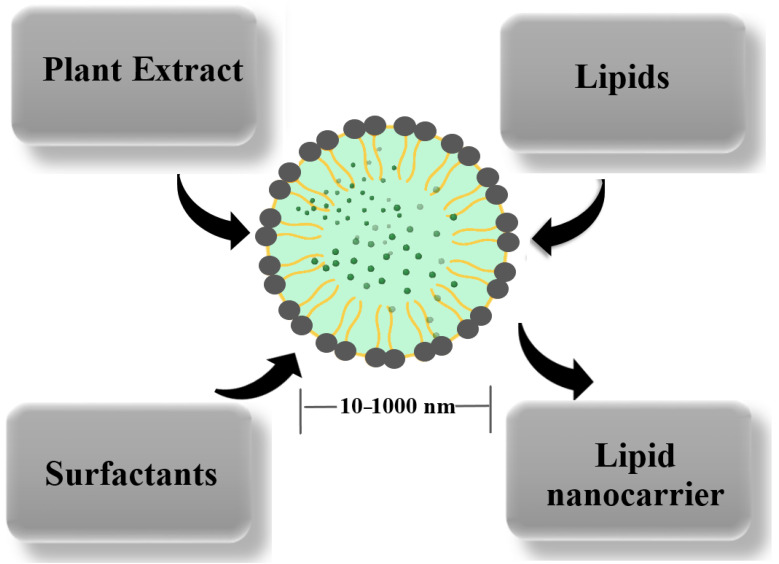
Obtaining lipid nanocarriers for antioxidant plant extracts’ encapsulation.

**Figure 7 pharmaceuticals-17-01738-f007:**
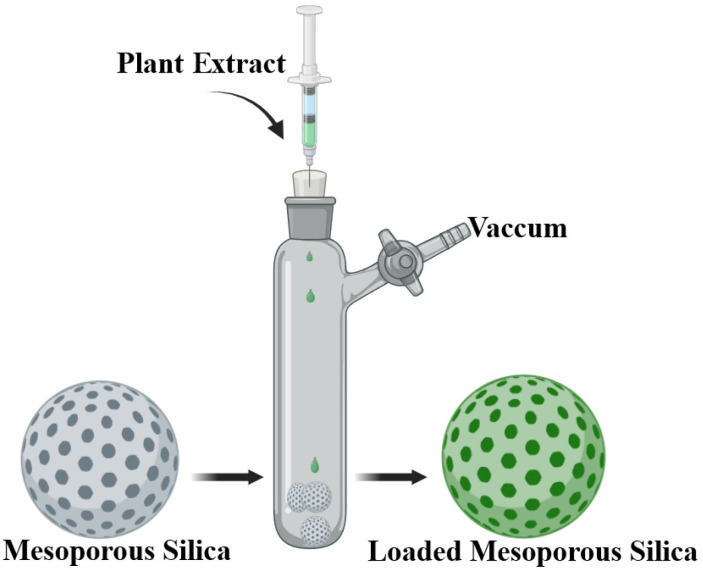
Vacuum-assisted loading method of plant extract into mesoporous silica [112].

**Figure 8 pharmaceuticals-17-01738-f008:**
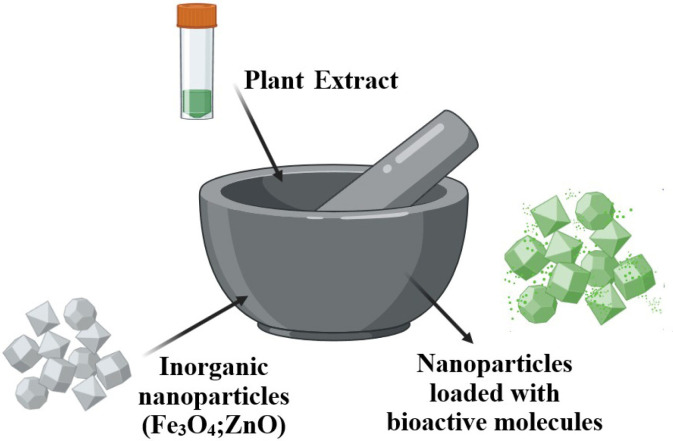
Antioxidant plant extract loaded onto nanoparticles by grinding in one-pot step.

**Table 1 pharmaceuticals-17-01738-t001:** Clinical trials demonstrating the role of antioxidant dietary supplements and gemmo-derivatives in metabolic and age-related conditions.

No.	Supplement	Dose	Results	Reference
1.	Supplement containing resveratrol	Oral intake, 500 mg, daily, 4 weeks	Reduced insulin resistance and decreased HDL cholesterol ratio in patients with diabetes mellitus and coronary heart disease	[95]
2.	Supplement containing antioxidant vitamins and minerals	Oral intake, 500 mg vitamin C, 400 IU vitamin E, 15 mg β-carotene, 80 mg zinc, daily, 12 months	Decreased progression of dry age-related macular degeneration (AMD), helping to preserve patient’s central vision	[96]
3.	Supplement containing xanthophyll pigments	Oral intake, 10 mg lutein, 2 mg zeaxanthin, daily, 36 months	Declined progression of vision loss in degenerative retinal patients of 50–60 years of age; provided better control of degenerative processes in patients over 60 years of age, compared to younger group	[97]
4.	*Corylus avellana* gemmo-derivative	Gavage administration, 6.711 µg/kg body weight, daily, 14 days	Reduced oxidative stress and liver damage in diabetic male adult mice with CCl_4_-induced liver fibrosis	[98]
5.	*Vaccinium myrtillus*, *Rosmarinus officinalis*, *Salix alba*, and *Ribes nigrum* gemmo-derivatives	Oral intake, 20 mL/kg body weight, daily, 14 days	Prophylactic phytotherapy in sub-acute toxicology studies conducted with white mice	[99]
6.	*Ribes nigrum* gemmo-derivative	Oral intake, 250 mL drink of 1:7500 diluted extract, daily, 4 weeks	Decreased microglia activation and TNF-α serum level in LPS-induced neuroinflammation in male Wistar rats	[100]

## Data Availability

Data are contained within this article.

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
