# Peer review of "The Role of Antioxidant Plant Extracts’ Composition and Encapsulation in Dietary Supplements and Gemmo-Derivatives, as Safe Adjuvants in Metabolic and Age-Related Conditions: A Review"

_pharmaceuticals, 2024, doi:10.3390/ph17121738_

Round 1
Reviewer 1 Report
Comments and Suggestions for Authors
The review primarily discusses the antioxidant properties of plant extracts and their encapsulation methods to enhance their efficacy in dietary supplements. These antioxidants are framed as potential adjuvants in managing metabolic and ophthalmic conditions, particularly by controlling oxidative stress and inflammation. The authors aim to present encapsulation technologies as a modern approach to boost the bioavailability and effectiveness of these extracts. The overall language style of the review is good and the logic is clear, but the citation and discussion of the references and the schematic diagram of the research summary need some revisions.
1. It is gemmo-derivatives not gemmoderivatives according to your reference 122.
2. It is NF-κB not NF-kB in the whole review.
3. Line 130 proposes that nutritional supplements include herbal products, isolated nutrients, dietary supplements, genetically modified foods, probiotics and prebiotics, enzymes, antioxidants, etc., but there are only three references here, and two of them are pharmacovigilance, which are not relevant to this sentence and need to be deleted. It is recommended that the author refer to DOI: 10.1021/acs.jafc.9b02520 DOI: 10.1002/fsn3.2586 DOI: 10.3390/ijms17060986 and cite the references reasonably.
4. More specific recommendations on establishing standardization can enhance the review's practical value.
5. While discussing mechanisms of action, further evidence from clinical studies on efficacy and safety can bolster claims, particularly for novel encapsulation methods.
6. Further depth on encapsulation protocols and parameters (such as particle size, loading efficiency) will make the discussion more robust.
7. Add a schematic diagram of the antioxidant mechanism of plant extracts: showing how the main antioxidant components in plant extracts (such as polyphenols and flavonoids) work by scavenging free radicals, inhibiting oxidative stress, etc.
8. Summarize the data of relevant clinical trials in the form of tables or charts, such as sample size, intervention methods, main findings, etc., to intuitively demonstrate the clinical application effects of plant extracts.
Reviewer 2 Report
Comments and Suggestions for Authors
- The review appears to be a superficial overview of the subject matter rather than a critical analysis of the literature.
- There is a lack of in-depth discussion of the scientific evidence supporting the claims made about the efficacy of the various products and techniques.
- The review does not critically evaluate the limitations and potential drawbacks of the research studies cited.

The language and style of the article are often unclear and difficult to follow.
Round 2
Reviewer 1 Report
Comments and Suggestions for Authors
The authors answered my questions well and I agree to accept this manuscript.
Reviewer 2 Report
Comments and Suggestions for Authors
The authors significantly revised the manuscript, corrected all comments, and gave a substantiated answer to all my questions. I think that the article has improved significantly and can be accepted for publication.
Comments on the Quality of English Language- Some sentences are too complex and could benefit from simpler structures to improve clarity.
- There are a few spelling errors and punctuation inconsistencies. Double-check these areas.